

# Integrative models of histopathological images and multi-omics data predict prognosis in endometrial carcinoma

Yueyi Li[1,*], Peixin Du[2,*], Hao Zeng[2], Yuhao Wei[3], Haoxuan Fu[4], Xi Zhong[5] and Xuelei Ma[1]

[1] Department of Targeting Therapy & Immunology, Cancer Center, West China Hospital, Sichuan University, Chengdu, Sichuan, China
[2] Laboratory of Integrative Medicine, Clinical Research Center for Breast, State Key Laboratory of Biotherapy, West China Hospital, Sichuan University, Chengdu, Sichuan, China
[3] West China School of Medicine, West China Hospital, Sichuan University, Chengdu, Sichuan, China
[4] Department of Statistics and Data Science, Wharton School, University of Pennsylvania, Philadelphia, Pennsylvania, United States of America
[5] Department of Critical Care Medicine, West China Hospital of Sichuan University, Chengdu, Sichuan, China
* These authors contributed equally to this work.

Corresponding authors
Xi Zhong, zhongxivip2006@163.com
Xuelei Ma, drmaxuelei@gmail.com

## ABSTRACT

**Objective:** This study aimed to predict the molecular features of endometrial carcinoma (EC) and the overall survival (OS) of EC patients using histopathological imaging.
**Methods:** The patients from The Cancer Genome Atlas (TCGA) were separated into the training set ($n$ = 215) and test set ($n$ = 214) in proportion of 1:1. By analyzing quantitative histological image features and setting up random forest model verified by cross-validation, we constructed prognostic models for OS. The model performance is evaluated with the time-dependent receiver operating characteristics (AUC) over the test set.
**Results:** Prognostic models based on histopathological imaging features (HIF) predicted OS in the test set (5-year AUC = 0.803). The performance of combining histopathology and omics transcends that of genomics, transcriptomics, or proteomics alone. Additionally, multi-dimensional omics data, including HIF, genomics, transcriptomics, and proteomics, attained the largest AUCs of 0.866, 0.869, and 0.856 at years 1, 3, and 5, respectively, showcasing the highest discrepancy in survival (HR = 18.347, 95% CI [11.09–25.65], $p$ < 0.001).
**Conclusions:** The results of this experiment indicated that the complementary features of HIF could improve the prognostic performance of EC patients. Moreover, the integration of HIF and multi-dimensional omics data might ameliorate survival prediction and risk stratification in clinical practice.

## INTRODUCTION

Endometrial carcinoma (EC) is the sixth most commonly encountered type of cancer in women and one of the most prevalent malignancies of the female reproductive system. Epidemiological data related that its incidence has risen globally in the last decades (*Fitzmaurice et al., 2018*), which could be partly attributed to the global epidemic of obesity (*Sheikh et al., 2014*; *Zeng et al., 2015*). Nevertheless, substantial geographical variations in cancer survival rates could also be observed between China and developed countries (*Zeng et al., 2018*). The main clinical research progress in EC was focused on sentinel lymph node mapping, adjuvant radiotherapy, and targeted therapy (*Garg, Jayaraj & Kumar, 2022*). However, despite of advances in EC treatment, its incidence and mortality rates continue to grow (*Xu et al., 2022*). Hence, identifying novel potential clinical prognostic biomarkers and therapeutic targets plays a vital role in enhancing survival outcomes in EC patients.

Biopsy and resection histopathological images are extensively used for cancer diagnosis, staging, prognosis, and treatment. With the takeoff of computer-aided image analysis systems, digital pathological images can now be examined more accurately, rapidly, and consistently than ever before. Moreover, they assist in reducing the pressure resulting from the shortage in manual evaluation (*Zhang et al., 2015*). Digital images are extracted with histopathological image features (HIF) comprising histological and morphological information that can be employed to determine the prognosis for numerous types of cancer, including lung cancer, breast cancer and prostate cancer (*Chen et al., 2015*; *Lee et al., 2017*; *Yu et al., 2016*). Given that tumor properties and its microenvironment are closely related to molecular alterations, some studies have utilized tumor morphology and gene expression to extract integrative prognostic signals (*Colen et al., 2014*; *Yuan et al., 2012*). The integration of histopathological imaging with genomics, transcriptomics and proteomics data in constructing prognostic models based on those high-dimensional features has been hypothesized to impart more predicting power in predicting the survival of cancer patients compared to using a single type of data (*Cheng et al., 2017*; *Zeng et al., 2020*). Indeed, it is feasible to achieve better superior accuracy with the implementation of multi-omics analysis.

The multi-omics model has been extensively applied in EC, and molecular indicators for EC have been incorporated into the NCCN Guidelines. Studies based on multi-omics and The Cancer Genome Atlas (TCGA) database divided EC into four distinct molecular subgroups: (1) polymerase epsilon (POLE) ultra-mutated group, (2) microsatellite instability (MSI)-hypermutated group, (3) copy number abnormalities-low (CN-L) group, and (4) copy number abnormalities-high (CN-H) group (*Kandoth et al., 2013*). In addition to these molecular subgroups, mutations are used to categorize EC, which can subsequently guide its treatment and prognosis. Patients with POLE-mutant EC typically have a favorable prognosis (*McConechy et al., 2016*). Noteworthily, mutations are crucial in predicting molecular subgroups. The integration of diverse information, such as genomics, transcriptomics, or any other known cancer-related information, might generate the ideal performance.

The goal of this research was to establish an image processing pipeline into which digital histopathological slides could be introduced; the system would then automatically extract their features and conduct sequential, systematic analyses to determine features correlated with those detected on the slides. This study also aimed to predict the mutations and molecular subtypes of EC using HIF and different machine-learning methods. To begin, the prognostic model was built using image feature to establish its robustness and reliability. Next, its performance was evaluated on the test set. Different multi-dimensional omics models were developed to compete for higher accuracy so that the survival risk could be more accurately predicted and to boost personalized medicines for EC patients.

## MATERIALS AND METHODS

### Data sources

The dataset in this study comprised genetic, transcriptional, and clinical information of EC patients from the TCGA database. Moreover, the corresponding protein profiles were collected from the Cancer Proteome Atlas (TCPA) repository. Variance-stabilizing transformation (VST) of the DESeq2 R package was applied to the mRNA sequencing data for normalization so as to extend the generality of the current model.

A total of 735 hematoxylin and eosin (H&E)-stained histopathological images of 465 patients were downloaded from The Cancer Imaging Archive (TCIA) to create the image features (Table S1). For adequate image feature recognition and feature extraction, all the histopathological tissue slides were formalin-fixed, and the slides were frozen to preserve cellular morphology. Each histopathological image was reviewed by experienced oncologists to ensure that the quality of the images was ideal for the ensuing analyses. A dataset of 465 samples was obtained; among them clinical, histopathological and multi-omics information was available for 429 samples.

### Feature engineering on the images

The flowchart of image segmentation and multi-omics model construction is illustrated in Fig. 1. The extremely high resolution of the original histopathological images made directly extracting their features intractable. Therefore, they were cropped into sub-images of 1,000 × 1,000 pixels using Openslide in Python. Thereafter, 60 sub-images from each original histopathological image were randomly selected for further analysis. CellProfiler was utilized to extract HIF from each sub-image and can transform the colored images of H&E-stained images into grayscale and extract useful features, including image intensity, correlation, area occupancy, image quality, granularity, object neighbors, object size shape and texture. Finally, the average value of 537 quantitative image features extracted from the 60 sub-images for each patient was calculated.

### Statistical analysis

Classifying molecular features: An attempt was made to predict the molecular features of EC using HIF. The data from the TCGA cohort were randomly split into a 1:1 ratio obtain a training set and a test set. Random splits maintain consistency at a positive rate. Several machine-learning algorithms were applied, including least absolute shrinkage and

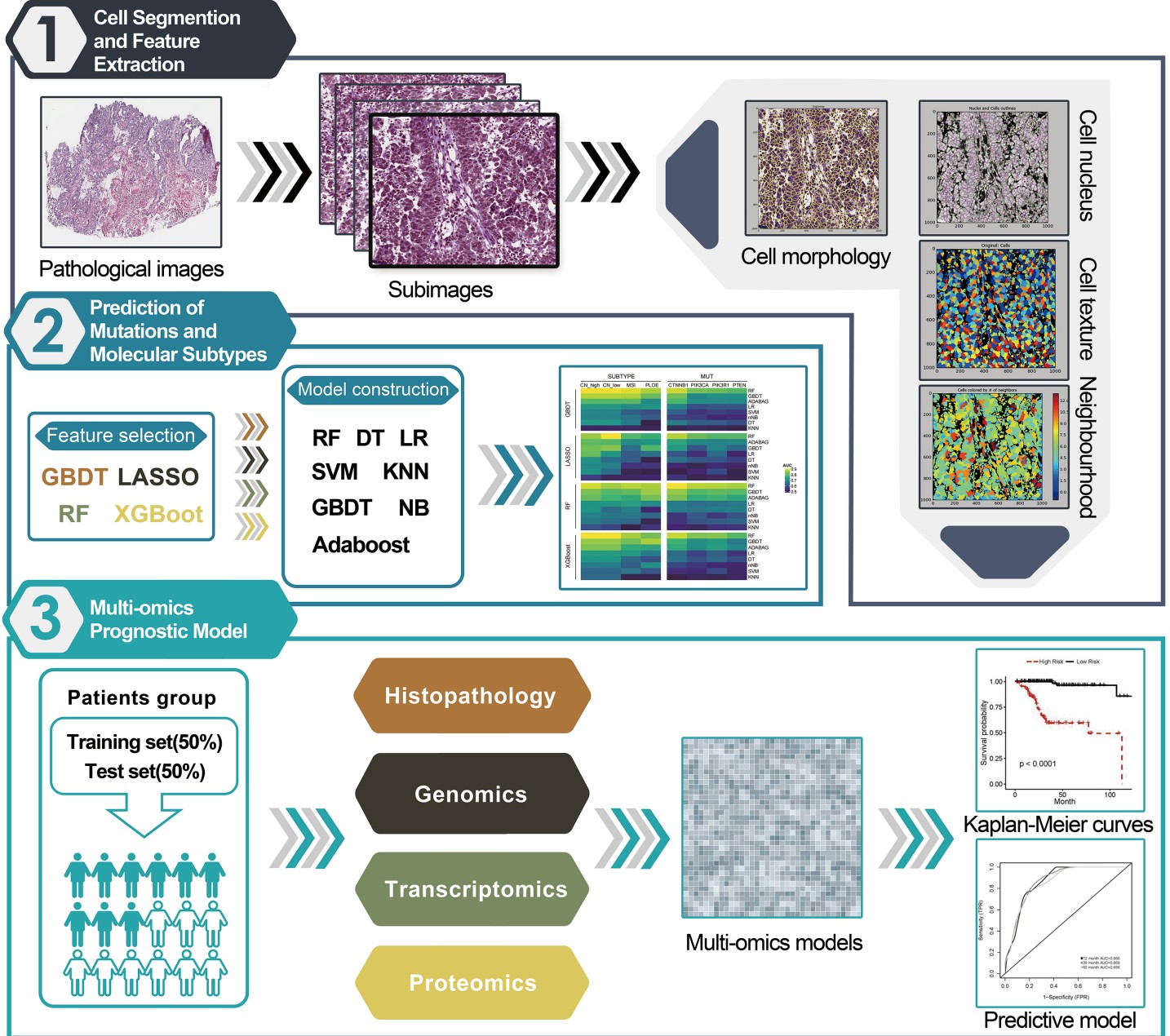

**Figure 1 Workflow chart for pipeline of data analysis and modeling.** (1) The histopathological images of EC 366 were cropped into smaller sub-images with size of 1,000 × 1,000 pixels. Next, CellProfiler was used to estimate the images and calculate the average values of various image features related to cell nucleus, morphology, and texture. (2) A total of 32 different combinations of machine learning algorithms were utilized for prediction of subtypes and mutations based on the extracted image information. (3) We incorporated features from histopathological images, genomics, transcriptomics, and proteomics to develop enhanced prognostic models using the random forest method. These models were trained on a specific dataset and evaluated for predictive performance using a separate test set.

selection operator (LASSO), gradient boosting decision tree (GBDT), extreme gradient boosting (XGBoost), and random forest (RF), to screen features in order of importance and subsequently narrow down the dimensionality of the data herein to 20 features. Next, RF, GBDT, ADABAG, logistic regression (LR), naive Bayes (NB), support vector machine

(SVM), decision tree (DT), and K-nearest neighbor (KNN) were selected to build classifiers based on the selected features to predict molecular substyles (CN-high, CN-low, MSI, and PLOE) and molecular mutations (CTNNB1, PIK3CA, PIK3R1, and PTEN). Finally, five-fold cross-validation was applied to fine tune the hyperparameter in the training set. Here, the area under the curve (AUC) of the receiver operating characteristic (ROC) curve served as evaluation metrics.

### Survival analysis

The TCGA cohort was divided into training and test sets. Patients in the training set were divided into high-expression and low-expression groups according to the median value of each histopathological imaging feature. Cox regression analysis was used to calculate the hazard ratio (HR) and the 95% confidence intervals of the overall survival (OS). Next, the Kaplan-Meier survival curve and the log-rank test were employed to estimate differences in survival outcomes between the two groups. $p < 0.05$ was considered as statistically significant.

### Data pre-processing

The genomic features and transcriptomics were screened to lower the dimensionality of data, and the top 100 most common somatic mutations in the training set were selected. One hundred of the most differently expressed genes (DEGs) of the training set patients, including the long-term (OS $\geq$ 60 months) and short-term (OS of 1–12 months at death) survival group, were chosen for survival prediction.

### Survival model development

The usefulness of different combinations of features, like single features (HIF, genomics, transcriptomics, proteomics) and the combination of two types of features (HIF + genomics, HIF + transcriptomics, HIF + proteomics, HIF + omics), in the prognostic model was evaluated and compared. Based on the HIF and genomic data, random survival forest (RSF) models were trained with five-fold cross-validation in the training set *via* the R randomForestSRC package to establish prognostic models. Moreover, patients were divided into high-risk and low-risk groups according to the median risk score estimated from the models. A time-dependent ROC curve was plotted to display the predictive capability of our prognostic model. Thereafter, the Kaplan-Meier method and log-rank test were applied to evaluate the survival difference between the two risk groups. Furthermore, to yield an in-depth understanding of the net clinical benefit of the models, decision curve analysis (DCA) was conducted with the threshold probabilities of each model based on a 5-year OS by the R "DCA" package.

## RESULTS

### Mutations and molecular subtypes prediction

Comparisons over different machine learning methods were performed on a 4 × 8 cross-test basis using four algorithms (GBDT, LASSO, RF, and XGBoost) for feature selection and eight algorithms (RF, GBDT, ADABAG, LR, NB, SVM, DT, and KNN) to build classifiers. The most outstanding performance was rendered by our RF model despite

the feature engineering process (Fig. 2A). Among 32 different combinations of possible classifiers, the performance of the RF model always peaked in the predicting process and also stood out among all feature filters, followed by XGBoost and GBDT. More specifically, the RF models accurately predicted common gene aberrations in EC, as reflected by their relatively high AUCs: CN-high (AUC = 0.826), CN-low (AUC = 0.882), MSI (AUC = 0.773), and PLOE (AUC = 0.754), and molecular mutations: CTNNB1 (AUC = 0.844), PIK3CA (AUC = 0.793), PIK3R1 (AUC = 0.773), PTEN l (AUC = 0.776) (Table S2). The results signaled that the HIF of EC could potentially be applied for predicting the molecular features and molecular subtypes of EC by statistical learning.

## Feature analysis of histopathological images for predicting prognosis

The training set was split into high-expression and low-expression groups depending on the median value of each HIF, and univariate Cox analyses were conducted to identify the prognostic value for OS (Table S3). The results uncovered that the top 20 HIF were correlated with OS ($p < 0.05$) (Fig. 2B). Among those histopathological features, the top four features with the smallest $p$-values were Median_Cells_Granularity_14, StDev_Cells_AreaShape_Orientation, StDev_Cells_Intensity_MassDisplacement, and StDev_Cells_Texture_DifferenceVariance_3_90 (Fig. 2C). The difference in OS between the high-expression and low-expression groups was significant, as demonstrated by the Kaplan-Meier survival curves and log-rank tests (Fig. 2C). Taken together, these analyses indicated that HIF of EC might be associated with prognosis.

Furthermore, patients were stratified into two subgroups based on survival time: the high-risk group (OS < 1 year) and the low-risk group (OS ≥ 5 years). Four features with discrepant categories showed up with their highest significance. They were further illustrated by high-expressed and low-expressed prognostic features (Fig. 2D). As displayed in Fig. 3, the histopathological sub-images from TCGA cohorts were from high-risk and low-risk groups. Preprocessing methods such as cell recognition and segmentation were subsequently conducted on the sub-images. In addition, algorithms were applied to identify and differentiate cell types within the images, aiming to validate the high accuracy of CellProfier for image processing.

## Histopathological image features and genomics integrated prognostic model

The RSF algorithm was included in our integrative model to validate the possibility of integrating histopathological and genomics data and to evaluate the value of genomics data in predicting prognosis. The 20 most common somatic mutations in the training set are presented in Fig. 4A. In addition, the time-dependent ROC curves of the model in the test set were analyzed. With our integrative prognostic model, the HIF + genomics model possessed the highest AUCs (0.841 at year 1, 0.829 at year 3, 0.815 at year 5) compared to the genomics model (0.765, 0.713, and 0.730) and HIF model (0.822, 0.805, and 0.803) at all time points (Figs. 4C–4E).

The test set was divided into a high-risk group (greater than the median) and a low-risk group (less than the median) according to the median value of the survival risk score

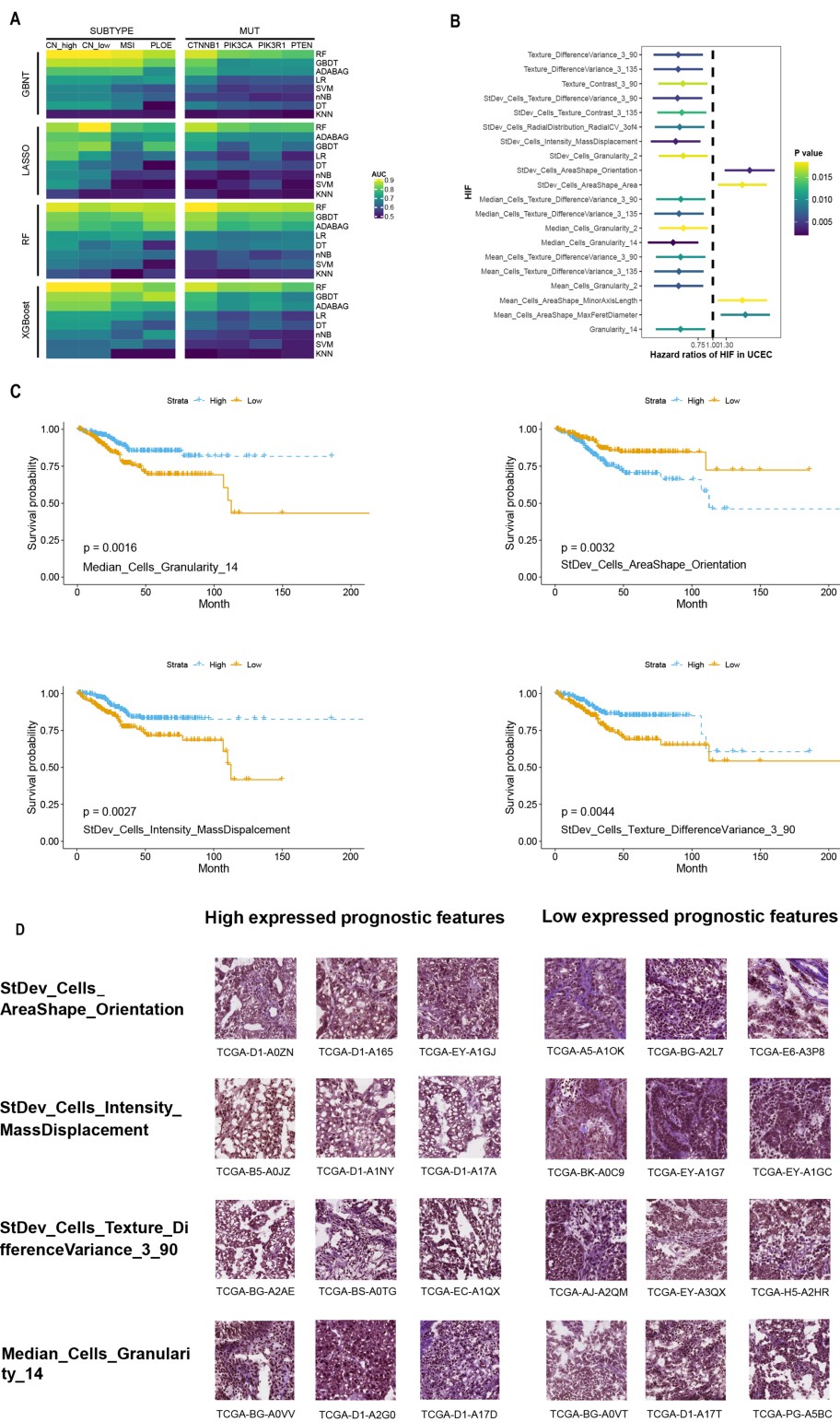

**Figure 2 Performance of prediction model using histopathological image features.** (A) Four algorithms (GBDT, LASSO, RF, XGBoost) were used for feature selection, while eight algorithms (RF, GBDT, ADABAG, LR, DT, SVM, nNB, KNN) function as prediction model in the training set. The models were evaluated on a separated set for demonstrating its prediction power. (B) Histopathological image features (HIF) that exhibited significant prognostic value ($p < 0.05$) in univariate

**Figure 2** (continued)
Cox analysis were identified. (C) Kaplan-Meier curves were plotted for specific features, namely "Median_Cells_Granularity_14", "StDev_Cells_AreaShape_Orientation", "StDev_Cells_Intensity_-MassDisplacement", and "StDev_Cells_Texture_DifferenceVariance_3_90". (D) Sub-images depicting patients with high and low expression of prognostic factors were generated. Each image feature was divided into groups based on its median value. 

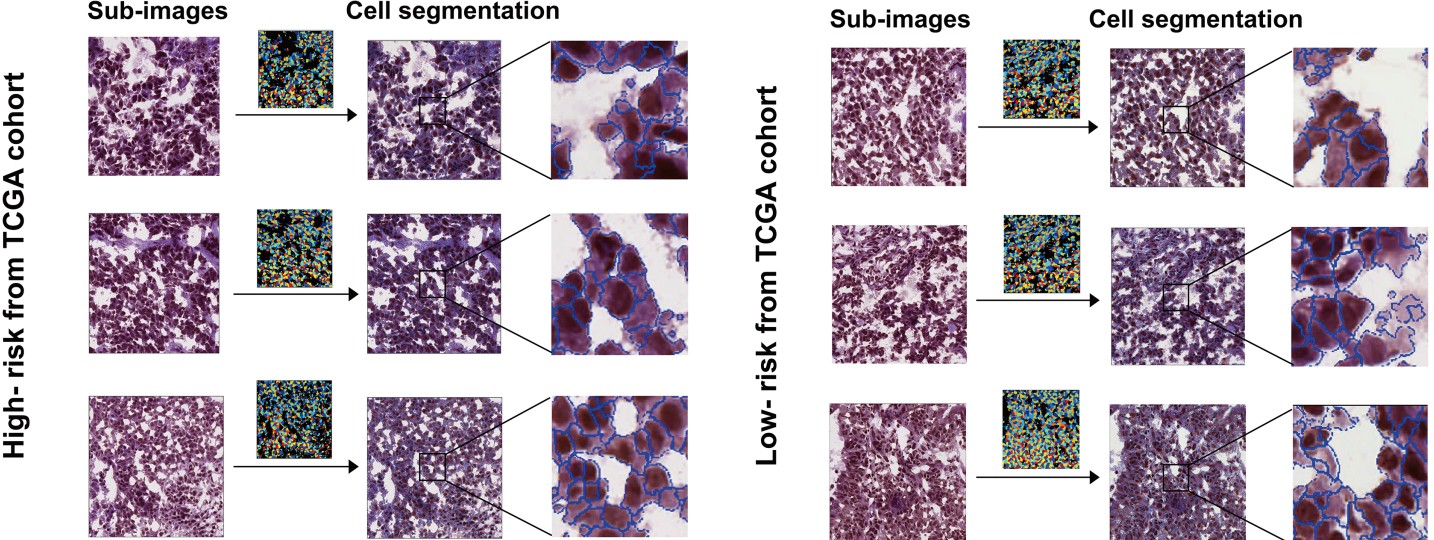

**Figure 3 Sub-images and processed images were generated for both the high-risk and low-risk groups, which were defined based on survival time.** The TCGA and TMA cohorts were partitioned into these two groups. 

predicted by the machine-learning models. Based on the Kaplan-Meier survival curves, our three models exhibited a satisfactory predictive value for the OS of in the high-risk and low-risk groups of the test set (Fig. 4B). Among our prognostic models, the integrative model achieved the best overall performance as it outperformed other models when differentiating the survival of patients. Compared to our integrative model, the HIF model achieved mundane results while the genomics model exhibited the worst performance in predicting survival. Lastly, the superiority of our integrative model demonstrated the efficacy of genomic and histopathological features in improving prognostic ability.

## Integrative model of histopathological images features and transcriptomics for predicting prognosis

Apart from genomics analysis, we combined histopathological data and transcriptomics data to investigate the predictive power of possible integration between HIF and mRNA transcription data of EC patients as well. Thus, the training set was stratified into short-term (OS of 1–12 months at death) and long-term survivors (OS ≥ 60 months). For mRNA sequencing data, the DEGs between the short-term and long-term groups were analyzed for dimension reduction. Gene Ontology (GO) enrichment analysis using Metascape (http://metascape.org) exposed that was closely associated with cell differentiation, anticoagulation, and blood circulation (Fig. 5A).

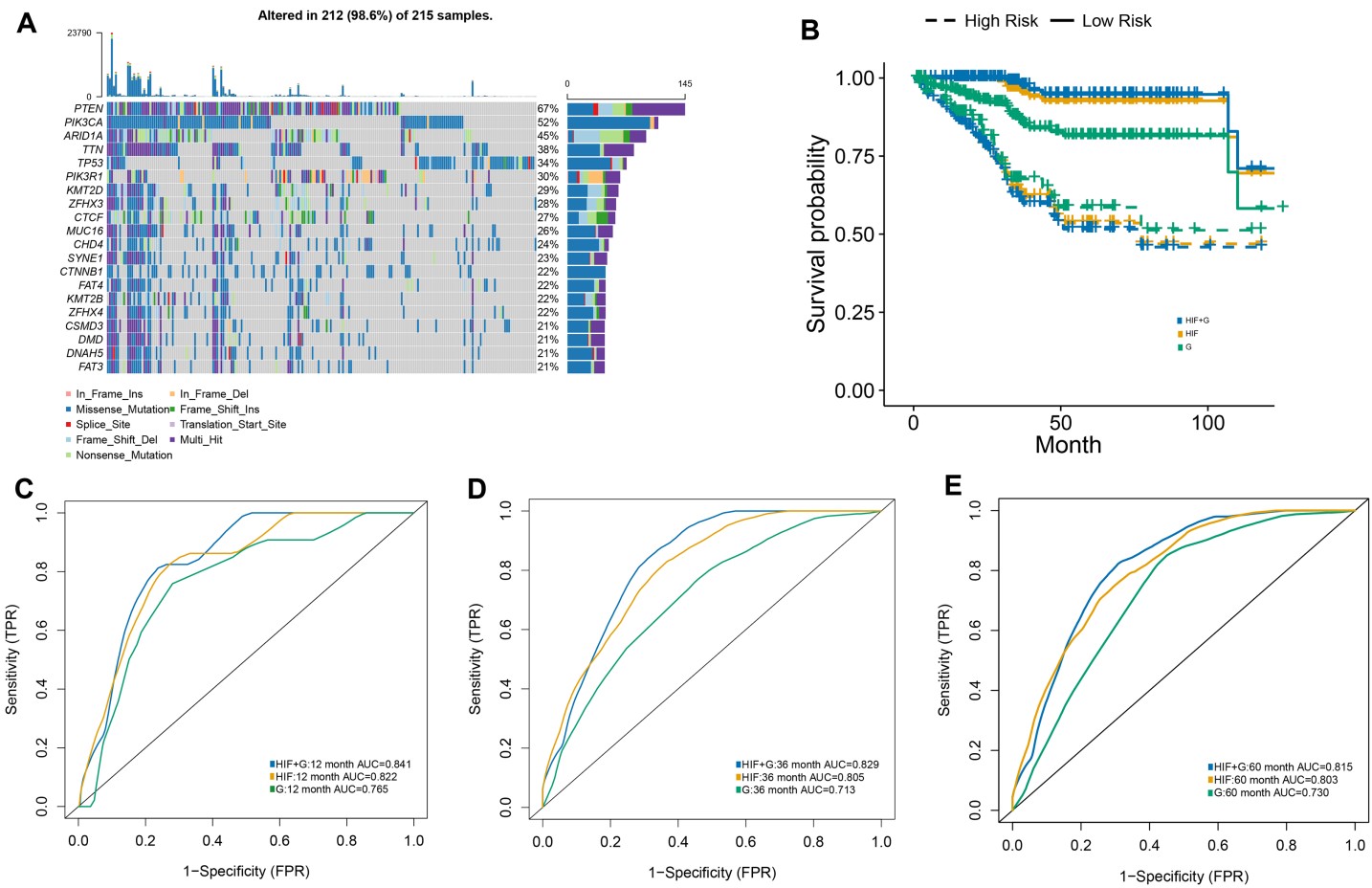

**Figure 4 The prognostic model used both histopathological image features and genomics.** (A) A waterfall plot was generated, illustrating the presence of 20 common somatic mutant genes within the training set. (B) The predictive performance of the integrative model, incorporating both histopathology image features and genomics (HIF + G), was compared to models using histopathological image features alone (HIF) or genomics alone (G). Kaplan-Meier survival curves were plotted, revealing that the HIF + G model exhibited superior predictive performance over HIF or G alone in the test set. (C–E) The area under the time-dependent receiver operating characteristic curve (AUC) calculated at 1, 3, and 5 years.

One hundred of the most significant DEGs were selected for modeling on the training set (Table S4). In the test set, HIF and transcriptomics (HIF + RNA) transcended the other two models with their superior predictive results (1-year AUC = 0.854, 3-year AUC = 0.844, 5-year AUC = 0.845) (Figs. 5C–5E). Furthermore, based on the HIF + RNA model, the prognosis for low-risk group was significantly higher than that in the high-risk group in the test set (Fig. 5B).

## A proteomics-integrated prognostic model of histopathological images

Besides the transcriptomic profile, the prognostic models were established using reverse phase protein array (RPPA) data with the inclusion of 203 proteins (Table S5). On the one hand, the predictive results obtained following the integration of image features and proteomics (HIF + P) outperformed those of the other two models in the test set (1-year

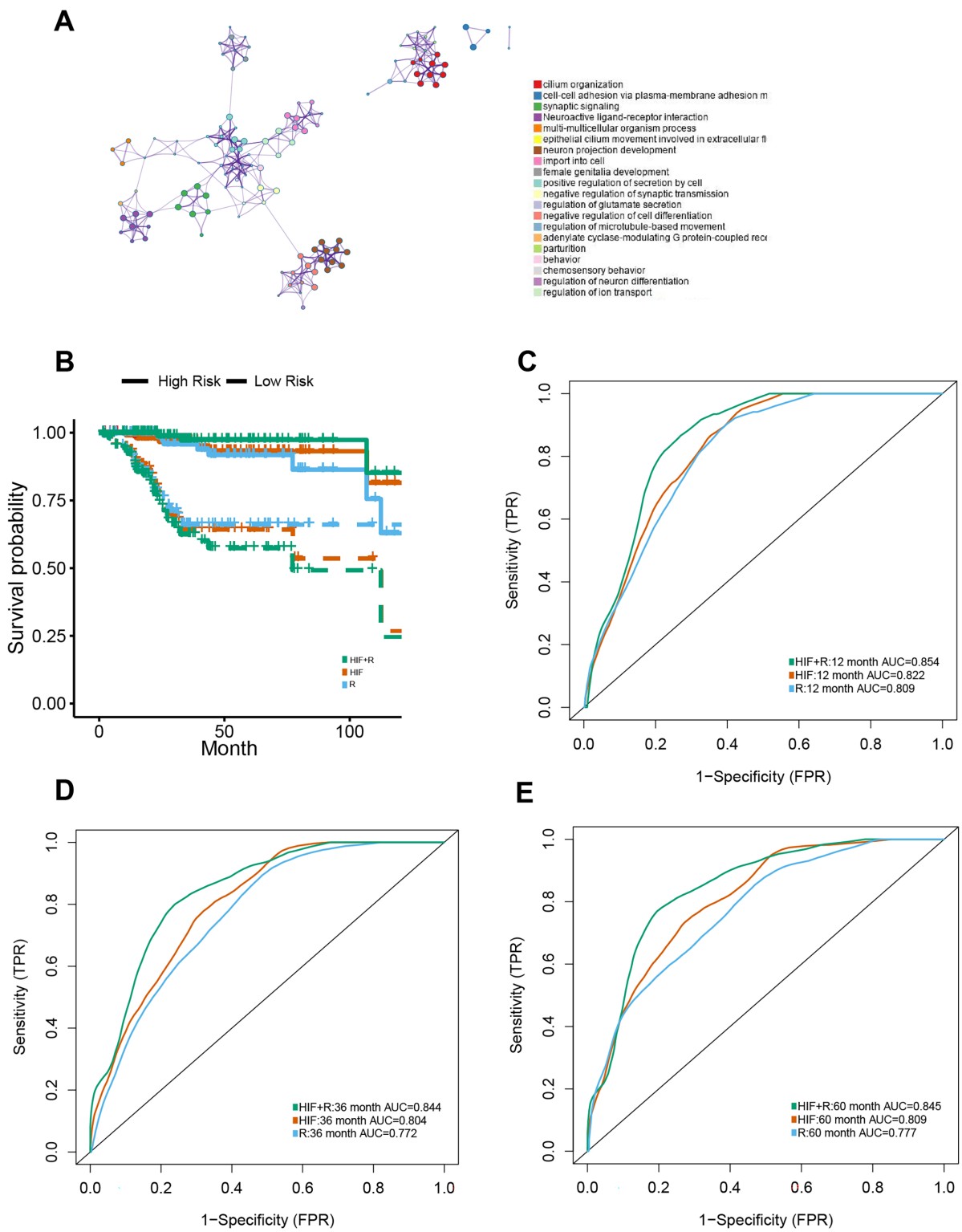

**Figure 5 Prognostic models constructed using histopathological image features (HIF) and transcriptomics (RNA).** (A) To understand the differentially expressed genes, a Gene Ontology enrichment network was built up, and the most enriched term was assigned to each cluster (refer to legends). (B) Demonstrated predictive performance of three models (HIF model, transcriptomics model, and HIF + RNA model) in the test set, showcased with Kaplan-Meier curves. (C–E) Area under the time-dependent ROC curves of three predictive models from test set.

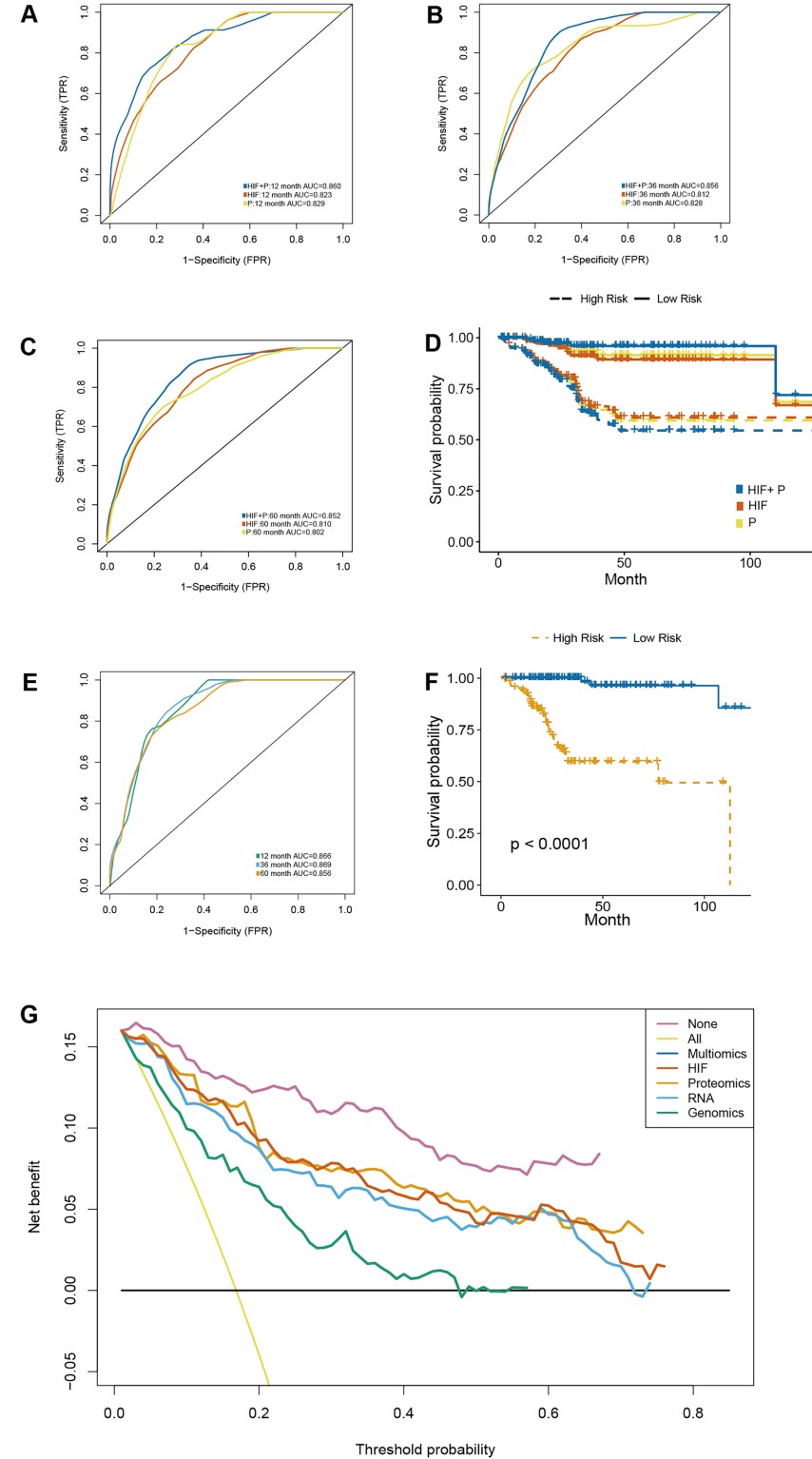

**Figure 6 A prognostic model developed by integrating histopathological image features with proteomics.** (A–C) The predictive power of histopathological image features (HIF), protein expression (P), and the combination of images and proteomics (HIF + P) for survival in the test set. (D) Kaplan-Meier curves demonstrated a more significant survival difference between high-risk and low-risk groups in the HIF + P model (E). The time-dependent receiver operating characteristic curve
**Figure 6 (continued)**
(F) and Kaplan-Meier curves for evaluation multi-omics model integrated image features, genomics, transcriptomics, and proteomics in the test set. (G) Decision curve analysis for each model in the test set, using an oblique line to represent the net benefit of intervening in all patients, and a horizontal line to represent the net benefit of no patients with intervention. The multi-omics model exhibited the highest net benefit compared to other models across a risk threshold range greater than 10%.

AUC = 0.860, 3-year AUC = 0.856, 5-year AUC = 0.852) (Figs. 6A–6C). On the other hand, Kaplan-Meier survival analyses of our established prognostic models, demonstrated significant differences in survival outcomes between the high-risk and low-risk patients (Fig. 6D), especially in the HIF + P model.

### Integrative multi-omics model for survival prediction

The previous analyses signaled that the combination of HIF and single omics (genomics, transcriptomics, and proteomics) had a better ability to predict the OS of EC patients. Therefore, the impact of integrating all the omics and histopathology image features in optimizing the results was assessed. In the test set, the AUC of OS at 1-year, 3-year, and 5-year was elevated to 0.866, 0.8969, and 0.856, respectively (Fig. 6E). Cox regression analysis and Kaplan-Meier curves indicated a significant difference in survival between the high-risk and low-risk groups (Fig. 6F) (HR = 18.347, 95% CI [11.09–25.65], $p < 0.001$) (Table S6). Additionally, decision curve analysis of the models was conducted on the test set, revealing that the multi-dimensional omics model had a greater net benefit than the others in the clinical decision-making process (Fig. 6G).

### DISCUSSION

Herein, the molecular subtypes, molecular mutations, and the prognosis of EC patients were predicted with HIF data, genomics data, transcription data, and proteomics data by applying image-processing and data analysis pipelines to extract HIF from histopathological images. Thereafter, molecular features were predicted with machine learning. HIF was subsequently used to predict the survival outcomes in EC patients. However, the predictive results from models using single-omics features or HIF were mediocre and far from ideal. Therefore, HIF and multi-omics information were integrated for patient survival prediction, with the expectation of superior overall performance. The integrated model was constructed to demonstrate that histomolecular integration could boost predictive performance. In other studies, several researchers predicted the therapeutic targets and different types of molecular mutations and features with multi-omics data (*Isobe et al., 2022*; *Parvathy Dharshini et al., 2022*; *Suter et al., 2022*). Moreover, multi-omics data was universally adopted for predicting the prognosis of different types of cancer, such as gastrointestinal cancers (*Li et al., 2022*; *Yuan et al., 2022*), high-grade serous ovarian cancer (*Sun et al., 2022*), lung adenocarcinoma (*Yang et al., 2022*) and head and neck squamous cell carcinoma (*Hildebrand et al., 2021*). To the best of our knowledge, this is the first study to predict the prognosis of EC patients with machine learning using multi-omics data.

Machine-learning models were established to predict CTNNB1, PIK3CA, PIK3R1, and PTEN mutations in EC patients with HIF. EC can be classified into several histological subtypes, which could be used to guide treatment decisions and determine the prognosis of EC patients (*Urick & Bell, 2019*). Meanwhile, PIK3CA mutation is one of the most common mutations in solid tumors, especially in endometrial (42–55%) (*Travaglino et al., 2022*); PIK3CA mutation is one of the most common mutations in solid tumors, especially in endometrial (42–55%) (*Kandoth et al., 2013*), cervical (42%) (*The Cancer Genome Atlas Research Network, 2017*), breast cancer (27–36%). The incidence of PIK3R1 (p85 α) mutations are higher in EC compared to any other cancer lineage, and the primary cause for triggering the PI3K pathway is the loss of PTEN protein (*Cheung et al., 2011*). Tumor cells might exhibit specific morphological features that indicate mutations in EC, such as necrosis (*Leskela et al., 2020*), mitotic index, transitional cell-like features, and pseudo-endometrioid (*Shia et al., 2008*). Quantitative image features were extracted instead of morphological features to yield a more scientific and stable method for predicting CTNNB1, PIK3CA, PIK3R1, and PTEN mutations. In the present study, histopathological features were used for the first time to accurately differentiate CTNNB1, PIK3CA, PIK3R1, and PTEN in EC patients. RF, with features selected by RF, outperformed all other models in the 32 combinations of machine-learning methods for predicting CTNNB1, PIK3CA, PIK3R1, and PTEN mutations in this analysis.

Simultaneously, the two types of EC were classified according to the four TCGA molecular subgroups of endometrial cancer (CN-high, CN-low, MSI, and POLE). Among the TCGA groups, the MSI group stands out with a predominating role, given that it represents great biological heterogeneity. The analysis demonstrated a correlation between molecular subgroups and patient outcomes, where the POLE ultra-mutated and copy number-high subgroups were associated with superior and unsatisfactory outcomes, respectively. In contrast, the microsatellite unstable hypermutated and copy number-low subgroups were linked to an intermediate outcome (*Loukovaara, Pasanen & Bützow, 2022*). In the current study, an attempt was made to develop a predictive model based on HIF to aid in the classification of molecular subtypes. Following the configuration of the machine-learning models for prediction, the empirical analytical results showed that the predictive model based on the random forest method demonstrated the best overall statistical performance. The MSI subgroup is typically detected by immunohistochemistry or genomic analysis. However, owing to cost and resource constraints, MSI testing has not been extensively used. Notably, recent studies have reported that AI can predict MSI from histopathological images (*Zhou et al., 2021*). The use of histopathological features in conjunction with machine-learning techniques could serve as a cost-effective and efficient tool to predict the molecular characteristics and subtypes of EC patients. Despite its promising potential, further comparison with clinical testing is still necessitated.

There is a marked difference in the prognosis of EC depending on the histological types (*Amant et al., 2005*). Therefore, contradictory disease mortality rates and prognoses make the histological type of EC an essential consideration regarding the health of women (*Sorosky, 2012*). In order to further investigate the prognosis of EC patients, a predictive model was developed based on HIF. Our optimal model (RF) achieved a remarkable

overall performance, which resulted in the AUC in the TCGA test set at 5 years being as high as 0.803. Even though the use of machine learning together with histopathological images for predicting the prognosis of lung and colon cancer patients has overwhelmed the field, related studies in EC are scarce (*Fremond et al., 2023*). In the present study, our model portrayed a stable predictive power in the test set, demonstrating the favorable generality of our model.

Since the occurrence and biological development of tumors are rather complex, multiple molecular-level data might help discover more features of tumors, which could subsequently lead to a better prognostic assessment and therapeutic intervention. This hypothesis led to the integration of different types of data in our models. Some researchers have coupled HIF with transcriptional data, resulting in significant improvement in EC survival predictions (*Salvesen et al., 2009*). In order to further improve the accuracy in predicting prognosis, our models were also constructed and tested according to different types of features, including HIF, genomics, transcriptomics, and proteomics data. Consequently, after training the machine-learning models on these data, it can be concluded that the predictive performance of a model using image features and one set of features is generally superior to that of a model using only one type of data. Moreover, the model combining multi-omic and imaging features was innovative. Astonishingly high AUC results from our multi-omics prognostic model insinuated that this model could be conducive to personalized risk stratification for EC patients.

Nevertheless, this study had some limitations that cannot be overlooked. To begin, the small sample size of positive cases of genetic variation and subtype limited the predictive accuracy of our models. Secondly, some confounding factors, such as alcohol consumption, complications, and chemotherapy, were not available in the TMA dataset; hence, the validity and generalizability of the combined prognostic model remained untested. Noteworthily, the multivariable Cox analysis demonstrated that although various interfering factors were present, the multi-group model could still deliver promising outcomes. Data on other interfering elements, such as optimal chemotherapy disassembly status and chemotherapy cycles that might have been associated with prognosis, were missing in the TCGA cohorts. Thus, it might be necessary to gather more information on confounding factors prior to assessing the prognostic power of the metabolomics models. Finally, the risk score threshold was based on the median value; for better patient stratification performance, the threshold of risk score should be tested more rigorously in large-scale studies.

## CONCLUSION

Collectively, this study put forth the great potential to predict genetic abnormalities, transcription subtypes, and survival outcomes in EC patients using histopathological features. In addition, the multi-omics predictive model combined with imaging, genomic, transcriptomic, and proteomic features could enhance the survival prediction of EC patients and assist in the prescription of individualized medicines for EC patients. Our model is anticipated to help clinicians assess the therapeutic options and prognosis of EC patients.

### Funding
The authors received no funding for this work.

### Competing Interests
The authors declare that they have no competing interests.

### Author Contributions

- Yueyi Li conceived and designed the experiments, performed the experiments, prepared figures and/or tables, authored or reviewed drafts of the article, and approved the final draft.
- Peixin Du conceived and designed the experiments, prepared figures and/or tables, and approved the final draft.
- Hao Zeng analyzed the data, authored or reviewed drafts of the article, and approved the final draft.
- Yuhao Wei performed the experiments, prepared figures and/or tables, and approved the final draft.
- Haoxuan Fu analyzed the data, authored or reviewed drafts of the article, and approved the final draft.
- Xi Zhong performed the experiments, authored or reviewed drafts of the article, and approved the final draft.
- Xuelei Ma conceived and designed the experiments, authored or reviewed drafts of the article, and approved the final draft.

### Data Availability
The code is available in the Supplemental Files.

The data is available in Zenodo: Xuelei Ma. (2023). UCEC-pathology images-1. https://doi.org/10.5281/zenodo.8103557.

Xuelei Ma. (2023). UCEC-pathology images-2. https://doi.org/10.5281/zenodo.8103838.

Xuelei Ma. (2023). UCEC-pathology images-3. https://doi.org/10.5281/zenodo.8103984.

Xuelei Ma. (2023). UCEC-pathology images-4. https://doi.org/10.5281/zenodo.8104233.

### Supplemental Information
Supplemental information for this article can be found online at http://dx.doi.org/10.7717/peerj.15674#supplemental-information.

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
