# Peer review of "Integrative models of histopathological images and multi-omics data predict prognosis in endometrial carcinoma"

_PeerJ, doi:10.7717/peerj.15674_

## Round 0.1 · original submission · Major Revisions

Authors should clarify whether or not the study included tumor adjacent tissues and control for confounding factors, as well as address limitations of the model. In addition, the data shown in Figure 6F does not appear to be credible: "In this analysis, the authors divided patients (more than 200 of them) into low risk and high risk groups. In the first 3-4 years, the death rate in the latter group declined steadily, by 40 percent, and then the curve turned sharply, flattening out over the next 3-4 years. Curves representing the previous group, on the other hand, remain absolutely flat for the first three or four years. Were it not for the sharp turn, one might have thought it was a great result." Please provide an explanation of the above data.

Reviewer 1 ·

Basic reporting

If I understand correctly, this study consists of two parts. In the first half, the authors try to establish an algorithm to predict molecular subtypes and/or the presence of specific gene abnormalities based on histological features. In the second half, they try to establish an algorithm to predict patient survival from the combination of histological features, genomic abnormalities, mRNA expression, and protein expression. All data used were publicly available.

The English of the manuscript is largely understandable but has room to be improved.

The authors claim “Though application of machine learning for predicting prognosis with histopathology images of lung and colon cancer has overwhelmed the field, but there are none related studies in EC”. A recent study describes a machine learning model to predict survival (via predicting molecular subtypes) from H&E stained slides (Fremond et al. 2023. “Interpretable Deep Learning Model to Predict the Molecular Classification of Endometrial Cancer from Haematoxylin and Eosin-Stained Whole-Slide Images: A Combined Analysis of the PORTEC Randomised Trials and Clinical Cohorts.” The Lancet. Digital Health 5 (2): e71–82.)

Some of the figures provided to this reviewer (embedded in pdf) are somewhat blurry.

Experimental design

Only to the Editor.

Validity of the findings

Only to the Editor.

·

Basic reporting

no comment

Experimental design

no comment

Validity of the findings

no comment

Additional comments

1. Omics sciences (i.e., genomics, transcriptomics, and proteomics), together with histopathological image features, are significant approaches for specific cancer characterization contributing to patient survival prediction; In the present manuscript, the authors considered a dataset of 429 Endometrial Carcinoma patients; prognostic models integrating histopathological image features with genomics, transcriptomics and proteomics were considered. According to the study data, the histopathological image features may act as an interesting prognostic biomarker in patients with EC and integration with omics sciences could help not only to the risk stratification but also to develop personalized therapies.
2. Limitation: Limited sample size; need of multi-center large-scale studies;
Strength: The Authors suggested an interesting approach for prediction and personalized therapies in case of EC. Moreover, the methodological approach may be also reproduced for other tumors.
3. When citing using the abbreviation 'Fig.', dots are missing after 'Fig' in some places of the manuscript.
4. The authors randomly selected 20 sub-images from the whole-slide images for each patient. Were the tumor adjacent regions included?
5. What about the external validity of this model, the authors require further analysis and discussion.

Reviewer 3 ·

Basic reporting

This manuscript submitted by Li et al. described the application of AI to decode histopathological image features which were used to predict molecular features. Moreover, the investigators attempted to use the combination with multi-dimensional omics data to predict overall survival in EC. The results are highly interesting and encouraging. The manuscript is well written and data clearly presented. However, there are a few major concerns that the authors need to address before its publication.
1. Ideally, adjacent regions should be able to provide more information thus enhance the efficacy of the models. For most of imagining model practice, they somehow include some adjacent regions for better performance, therefore it is generally suggested to include the adjacent regions. In this paper, were the tumor adjacent regions included, and if not, why they were not included?
2. The random nature of the train test split give rise to the possibility of accidental ideal performance. To avoid such “surprise”, one must carefully design the train test split schema by using techniques like stratified train test split. Can author provide more information on how train test split is performed in this paper?
3. There are piles of different possible criteria available for usage in feature selection, like SHAP score and feature importance ranking. Author should illustrate the specific method used for feature selection, and along with the way dealing with the confounders.
4. Authors seem to overemphasize the performance of the model but omit to talk more about the limitations. AI for medicine by now are under strict supervision and the expansion is yet await to happen, can authors provide more insights on possible drawbacks of the model which obstacle it from further applications?

Experimental design

no

Validity of the findings

no

Additional comments

no

---

## Round 0.2 · accepted · Accept

The authors have addressed the reviewers' concerns.